# DIAGNOSING AND ENHANCING VAE MODELS

**Bin Dai**
Institute for Advanced Study
Tsinghua University
Beijing, China
daib13@mails.tsinghua.edu.cn

**David Wipf**
Microsoft Research
Beijing, China
davidwipf@gmail.com

## ABSTRACT

Although variational autoencoders (VAEs) represent a widely influential deep generative model, many aspects of the underlying energy function remain poorly understood. In particular, it is commonly believed that Gaussian encoder/decoder assumptions reduce the effectiveness of VAEs in generating realistic samples. In this regard, we rigorously analyze the VAE objective, differentiating situations where this belief is and is not actually true. We then leverage the corresponding insights to develop a simple VAE enhancement that requires no additional hyperparameters or sensitive tuning. Quantitatively, this proposal produces crisp samples and stable FID scores that are actually competitive with a variety of GAN models, all while retaining desirable attributes of the original VAE architecture. The code for our model is available at https://github.com/daib13/TwoStageVAE.

## 1 INTRODUCTION

Our starting point is the desire to learn a probabilistic generative model of observable variables $\boldsymbol{x} \in \chi$, where $\chi$ is a $r$-dimensional manifold embedded in $\mathbb{R}^d$. Note that if $r = d$, then this assumption places no restriction on the distribution of $\boldsymbol{x} \in \mathbb{R}^d$ whatsoever; however, the added formalism is introduced to handle the frequently encountered case where $\boldsymbol{x}$ possesses low-dimensional structure relative to a high-dimensional ambient space, i.e., $r \ll d$. In fact, the very utility of generative models of continuous data, and their attendant low-dimensional representations, often hinges on this assumption (Bengio et al., 2013). It therefore behooves us to explicitly account for this situation.

Beyond this, we assume that $\chi$ is a simple Riemannian manifold, which means there exists a diffeomorphism $\varphi$ between $\chi$ and $\mathbb{R}^r$, or more explicitly, the mapping $\varphi : \chi \mapsto \mathbb{R}^r$ is invertible and differentiable. Denote a ground-truth probability measure on $\chi$ as $\mu_{gt}$ such that the probability mass of an infinitesimal $d\boldsymbol{x}$ on the manifold is $\mu_{gt}(d\boldsymbol{x})$ and $\int_\chi \mu_{gt}(d\boldsymbol{x}) = 1$.

The variational autoencoder (VAE) (Kingma & Welling, 2014; Rezende et al., 2014) attempts to approximate this ground-truth measure using a parameterized density $p_\theta(\boldsymbol{x})$ defined across all of $\mathbb{R}^d$ since any underlying generative manifold is unknown in advance. This density is further assumed to admit the latent decomposition $p_\theta(\boldsymbol{x}) = \int p_\theta(\boldsymbol{x}|\boldsymbol{z})p(\boldsymbol{z})d\boldsymbol{z}$, where $\boldsymbol{z} \in \mathbb{R}^\kappa$ serves as a low-dimensional representation, with $\kappa \approx r$ and prior $p(\boldsymbol{z}) = \mathcal{N}(\boldsymbol{z}|\boldsymbol{0}, \boldsymbol{I})$.

Ideally we might like to minimize the negative log-likelihood $-\log p_\theta(\boldsymbol{x})$ averaged across the ground-truth measure $\mu_{gt}$, i.e., solve $\min_\theta \int_\chi -\log p_\theta(\boldsymbol{x})\mu_{gt}(d\boldsymbol{x})$. Unfortunately though, the required marginalization over $\boldsymbol{z}$ is generally infeasible. Instead the VAE model relies on tractable *encoder* $q_\phi(\boldsymbol{z}|\boldsymbol{x})$ and *decoder* $p_\theta(\boldsymbol{x}|\boldsymbol{z})$ distributions, where $\phi$ represents additional trainable parameters. The canonical VAE cost is a bound on the average negative log-likelihood given by

$$\mathcal{L}(\theta, \phi) \triangleq \int_\chi \left\{ -\log p_\theta(\boldsymbol{x}) + \mathbb{KL}\left[q_\phi(\boldsymbol{z}|\boldsymbol{x})||p_\theta(\boldsymbol{z}|\boldsymbol{x})\right] \right\} \mu_{gt}(d\boldsymbol{x}) \geq \int_\chi -\log p_\theta(\boldsymbol{x})\mu_{gt}(d\boldsymbol{x}), \quad (1)$$

where the inequality follows directly from the non-negativity of the KL-divergence. Here $\phi$ can be viewed as tuning the tightness of bound, while $\theta$ dictates the actual estimation of $\mu_{gt}$. Using a few standard manipulations, this bound can also be expressed as

$$\mathcal{L}(\theta, \phi) = \int_\chi \left\{ -\mathbb{E}_{q_\phi(\boldsymbol{z}|\boldsymbol{x})}\left[\log p_\theta(\boldsymbol{x}|\boldsymbol{z})\right] + \mathbb{KL}\left[q_\phi(\boldsymbol{z}|\boldsymbol{x})||p(\boldsymbol{z})\right] \right\} \mu_{gt}(d\boldsymbol{x}), \quad (2)$$

which explicitly involves the encoder/decoder distributions and is conveniently amenable to SGD optimization of $\{\theta, \phi\}$ via a reparameterization trick (Kingma & Welling, 2014; Rezende et al.,

2014). The first term in (2) can be viewed as a reconstruction cost (or a stochastic analog of a traditional autoencoder), while the second penalizes posterior deviations from the prior $p(z)$. Additionally, for any realizable implementation via SGD, the integration over $\chi$ must be approximated via a finite sum across training samples $\{x^{(i)}\}_{i=1}^{n}$ drawn from $\mu_{gt}$. Nonetheless, examining the true objective $\mathcal{L}(\theta, \phi)$ can lead to important, practically-relevant insights.

At least in principle, $q_\phi(z|x)$ and $p_\theta(x|z)$ can be arbitrary distributions, in which case we could simply enforce $q_\phi(z|x) = p_\theta(z|x) \propto p_\theta(x|z)p(z)$ such that the bound from (1) is tight. Unfortunately though, this is essentially always an intractable undertaking. Consequently, largely to facilitate practical implementation, a commonly adopted distributional assumption for continuous data is that both $q_\phi(z|x)$ and $p_\theta(x|z)$ are Gaussian. This design choice has previously been cited as a key limitation of VAEs (Burda et al., 2015; Kingma et al., 2016), and existing quantitative tests of generative modeling quality thus far dramatically favor contemporary alternatives such as generative adversarial networks (GAN) (Goodfellow et al., 2014). Regardless, because the VAE possesses certain desirable properties relative to GAN models (e.g., stable training (Tolstikhin et al., 2018), interpretable encoder/inference network (Brock et al., 2016), outlier-robustness (Dai et al., 2018), etc.), it remains a highly influential paradigm worthy of examination and enhancement.

In Section 2 we closely investigate the implications of VAE Gaussian assumptions leading to a number of interesting diagnostic conclusions. In particular, we differentiate the situation where $r = d$, in which case we prove that recovering the ground-truth distribution is actually possible iff the VAE global optimum is reached, and $r < d$, in which case the VAE global optimum can be reached by solutions that reflect the ground-truth distribution almost everywhere, but not necessarily uniquely so. In other words, there could exist alternative solutions that both reach the global optimum and yet do not assign the same probability measure as $\mu_{gt}$.

Section 3 then further probes this non-uniqueness issue by inspecting necessary conditions of global optima when $r < d$. This analysis reveals that an optimal VAE parameterization will provide an encoder/decoder pair capable of perfectly reconstructing all $x \in \chi$ using *any* $z$ drawn from $q_\phi(z|x)$. Moreover, we demonstrate that the VAE accomplishes this using a degenerate latent code whereby only $r$ dimensions are effectively active. Collectively, these results indicate that the VAE global optimum can in fact uniquely learn a mapping to the correct ground-truth manifold when $r < d$, but not necessarily the correct probability measure *within* this manifold, a critical distinction.

Next we leverage these analytical results in Section 4 to motivate an almost trivially-simple, two-stage VAE enhancement for addressing typical regimes when $r < d$. In brief, the first stage just learns the manifold per the allowances from Section 3, and in doing so, provides a mapping to a lower dimensional intermediate representation with no degenerate dimensions that mirrors the $r = d$ regime. The second (much smaller) stage then only needs to learn the correct probability measure on this intermediate representation, which is possible per the analysis from Section 2. Experiments from Sections 5 and 6 empirically corroborate motivational theory and reveal that the proposed two-stage procedure can generate high-quality samples, reducing the blurriness often attributed to VAE models in the past (Dosovitskiy & Brox, 2016; Larsen et al., 2015). And to the best of our knowledge, this is the first demonstration of a VAE pipeline that can produce stable FID scores, an influential recent metric for evaluating generated sample quality (Heusel et al., 2017), that are comparable to GAN models under neutral testing conditions. Moreover, this is accomplished without additional penalties, cost function modifications, or sensitive tuning parameters. Finally, an extended version of this work can be found in (Dai & Wipf, 2019). There we include additional results, consideration of disentangled representations, as well as a comparative discussion of broader VAE modeling paradigms such as those involving normalizing flows or parameterized families for $p(z)$.

## 2   HIGH-LEVEL IMPACT OF VAE GAUSSIAN ASSUMPTIONS

Conventional wisdom suggests that VAE Gaussian assumptions will introduce a gap between $\mathcal{L}(\theta, \phi)$ and the ideal negative log-likelihood $\int_\chi -\log p_\theta(x)\mu_{gt}(dx)$, compromising efforts to learn the ground-truth measure. However, we will now argue that this pessimism is in some sense premature. In fact, we will demonstrate that, even with the stated Gaussian distributions, there exist parameters $\phi$ and $\theta$ that can simultaneously: (*i*) Globally optimize the VAE objective and, (*ii*) Recover the ground-truth probability measure in a certain sense described below. This is possible because, at least for some coordinated values of $\phi$ and $\theta$, $q_\phi(z|x)$ and $p_\theta(z|x)$ can indeed become

arbitrarily close. Before presenting the details, we first formalize a $\kappa$-simple VAE, which is merely a VAE model with explicit Gaussian assumptions and parameterizations:

**Definition 1** *A $\kappa$-simple VAE is defined as a VAE model with $dim[\boldsymbol{z}] = \kappa$ latent dimensions, the Gaussian encoder $q_\phi(\boldsymbol{z}|\boldsymbol{x}) = \mathcal{N}(\boldsymbol{z}|\boldsymbol{\mu}_z, \boldsymbol{\Sigma}_z)$, and the Gaussian decoder $p_\theta(\boldsymbol{x}|\boldsymbol{z}) = \mathcal{N}(\boldsymbol{x}|\boldsymbol{\mu}_x, \boldsymbol{\Sigma}_x)$. Moreover, the encoder moments are defined as $\boldsymbol{\mu}_z = f_{\mu_z}(\boldsymbol{x}; \phi)$ and $\boldsymbol{\Sigma}_z = \boldsymbol{S}_z \boldsymbol{S}_z^\top$ with $\boldsymbol{S}_z = f_{S_z}(\boldsymbol{x}; \phi)$. Likewise, the decoder moments are $\boldsymbol{\mu}_x = f_{\mu_x}(\boldsymbol{z}; \theta)$ and $\boldsymbol{\Sigma}_x = \gamma \boldsymbol{I}$. Here $\gamma > 0$ is a tunable scalar, while $f_{\mu_z}$, $f_{S_z}$ and $f_{\mu_x}$ specify parameterized differentiable functional forms that can be arbitrarily complex, e.g., a deep neural network.*

Equipped with these definitions, we will now demonstrate that a $\kappa$-simple VAE, with $\kappa \geq r$, can achieve the optimality criteria (*i*) and (*ii*) from above. In doing so, we first consider the simpler case where $r = d$, followed by the extended scenario with $r < d$. The distinction between these two cases turns out to be significant, with practical implications to be explored in Section 4.

## 2.1 MANIFOLD DIMENSION EQUAL TO AMBIENT SPACE DIMENSION ($r = d$)

We first analyze the specialized situation where $r = d$. Assuming $p_{gt}(\boldsymbol{x}) \triangleq \mu_{gt}(d\boldsymbol{x})/d\boldsymbol{x}$ exists everywhere in $\mathbb{R}^d$, then $p_{gt}(\boldsymbol{x})$ represents the ground-truth probability density with respect to the standard Lebesgue measure in Euclidean space. Given these considerations, the minimal possible value of (1) will necessarily occur if

$$\mathbb{KL}\left[q_\phi(\boldsymbol{z}|\boldsymbol{x})||p_\theta(\boldsymbol{z}|\boldsymbol{x})\right] = 0 \quad \text{and} \quad p_\theta(\boldsymbol{x}) = p_{gt}(\boldsymbol{x}) \text{ almost everywhere.} \tag{3}$$

This follows because by VAE design it must be that $\mathcal{L}(\theta, \phi) \geq -\int p_{gt}(\boldsymbol{x}) \log p_{gt}(\boldsymbol{x})d\boldsymbol{x}$, and in the present context, this lower bound is achievable iff the conditions from (3) hold. Collectively, this implies that the approximate posterior produced by the encoder $q_\phi(\boldsymbol{z}|\boldsymbol{x})$ is in fact perfectly matched to the actual posterior $p_\theta(\boldsymbol{z}|\boldsymbol{x})$, while the corresponding marginalized data distribution $p_\theta(\boldsymbol{x})$ is perfectly matched the ground-truth density $p_{gt}(\boldsymbol{x})$ as desired. Perhaps surprisingly, a $\kappa$-simple VAE can actually achieve such a solution:

**Theorem 1** *Suppose that $r = d$ and there exists a density $p_{gt}(\boldsymbol{x})$ associated with the ground-truth measure $\mu_{gt}$ that is nonzero everywhere on $\mathbb{R}^d$.[1] Then for any $\kappa \geq r$, there is a sequence of $\kappa$-simple VAE model parameters $\{\theta_t^*, \phi_t^*\}$ such that*

$$\lim_{t \to \infty} \mathbb{KL}\left[q_{\phi_t^*}(\boldsymbol{z}|\boldsymbol{x})||p_{\theta_t^*}(\boldsymbol{z}|\boldsymbol{x})\right] = 0 \quad \text{and} \quad \lim_{t \to \infty} p_{\theta_t^*}(\boldsymbol{x}) = p_{gt}(\boldsymbol{x}) \text{ almost everywhere.} \tag{4}$$

All the proofs can be found in (Dai & Wipf, 2019). So at least when $r = d$, the VAE Gaussian assumptions need not actually prevent the optimal ground-truth probability measure from being recovered, as long as the latent dimension is sufficiently large (i.e., $\kappa \geq r$). And contrary to popular notions, a richer class of distributions is not required to achieve this. Of course Theorem 1 only applies to a restricted case that excludes $d > r$; however, later we will demonstrate that a key consequence of this result can nonetheless be leveraged to dramatically enhance VAE performance.

## 2.2 MANIFOLD DIMENSION LESS THAN AMBIENT SPACE DIMENSION ($r < d$)

When $r < d$, additional subtleties are introduced that will be unpacked both here and in the sequel. To begin, if both $q_\phi(\boldsymbol{z}|\boldsymbol{x})$ and $p_\theta(\boldsymbol{x}|\boldsymbol{z})$ are arbitrary/unconstrained (i.e., not necessarily Gaussian), then $\inf_{\phi, \theta} \mathcal{L}(\theta, \phi) = -\infty$. To achieve this global optimum, we need only choose $\phi$ such that $q_\phi(\boldsymbol{z}|\boldsymbol{x}) = p_\theta(\boldsymbol{z}|\boldsymbol{x})$ (minimizing the KL term from (1)) while selecting $\theta$ such that all probability mass collapses to the correct manifold $\boldsymbol{\chi}$. In this scenario the density $p_\theta(\boldsymbol{x})$ will become unbounded on $\boldsymbol{\chi}$ and zero elsewhere, such that $\int_{\boldsymbol{\chi}} -\log p_\theta(\boldsymbol{x})\mu_{gt}(d\boldsymbol{x})$ will approach negative infinity.

But of course the stated Gaussian assumptions from the $\kappa$-simple VAE model could ostensibly prevent this from occurring by causing the KL term to blow up, counteracting the negative log-likelihood factor. We will now analyze this case to demonstrate that this need not happen. Before

---

[1]This nonzero assumption can be replaced with a much looser condition. Specifically, if there exists a diffeomorphism between the set $\{\boldsymbol{x}|p_{gt}(\boldsymbol{x}) \neq 0\}$ and $\mathbb{R}^d$, then it can be shown that Theorem 1 still holds even if $p_{gt}(\boldsymbol{x}) = 0$ for some $\boldsymbol{x} \in \mathbb{R}^d$.

proceeding to this result, we first define a manifold density $\tilde{p}_{gt}(\boldsymbol{x})$ as the probability density (assuming it exists) of $\mu_{gt}$ with respect to the volume measure of the manifold $\boldsymbol{\chi}$. If $d = r$ then this volume measure reduces to the standard Lebesgue measure in $\mathbb{R}^d$ and $\tilde{p}_{gt}(\boldsymbol{x}) = p_{gt}(\boldsymbol{x})$; however, when $d > r$ a density $p_{gt}(\boldsymbol{x})$ defined in $\mathbb{R}^d$ will not technically exist, while $\tilde{p}_{gt}(\boldsymbol{x})$ is still perfectly well-defined. We then have the following:

**Theorem 2** *Assume $r < d$ and that there exists a manifold density $\tilde{p}_{gt}(\boldsymbol{x})$ associated with the ground-truth measure $\mu_{gt}$ that is nonzero everywhere on $\boldsymbol{\chi}$. Then for any $\kappa \geq r$, there is a sequence of $\kappa$-simple VAE model parameters $\{\theta_t^*, \phi_t^*\}$ such that*

$$(i) \quad \lim_{t \to \infty} \mathbb{KL}\left[q_{\phi_t^*}(\boldsymbol{z}|\boldsymbol{x})||p_{\theta_t^*}(\boldsymbol{z}|\boldsymbol{x})\right] = 0 \quad and \quad \lim_{t \to \infty} \int_{\boldsymbol{\chi}} -\log p_{\theta_t^*}(\boldsymbol{x})\mu_{gt}(d\boldsymbol{x}) = -\infty, \quad (5)$$

$$(ii) \quad \lim_{t \to \infty} \int_{\boldsymbol{x} \in A} p_{\theta_t^*}(\boldsymbol{x})d\boldsymbol{x} = \mu_{gt}(A \cap \boldsymbol{\chi}) \quad (6)$$
*for all measurable sets $A \subseteq \mathbb{R}^d$ with $\mu_{gt}(\partial A \cap \boldsymbol{\chi}) = 0$, where $\partial A$ is the boundary of $A$.*

Technical details notwithstanding, Theorem 2 admits a very intuitive interpretation. First, (5) directly implies that the VAE Gaussian assumptions do not prevent minimization of $\mathcal{L}(\theta, \phi)$ from converging to minus infinity, which can be trivially viewed as a globally optimum solution. Furthermore, based on (6), this solution can be achieved with a limiting density estimate that will assign a probability mass to most all measurable subsets of $\mathbb{R}^d$ that is indistinguishable from the ground-truth measure (which confines all mass to $\boldsymbol{\chi}$). Hence this solution is more-or-less an arbitrarily-good approximation to $\mu_{gt}$ for all practical purposes.[2]

Regardless, there is an absolutely crucial distinction between Theorem 2 and the simpler case quantified by Theorem 1. Although both describe conditions whereby the $\kappa$-simple VAE can achieve the minimal possible objective, in the $r = d$ case achieving the lower bound (whether the specific parameterization for doing so is unique or not) necessitates that the ground-truth probability measure has been recovered almost everywhere. But the $r < d$ situation is quite different because we have not ruled out the possibility that a different set of parameters $\{\theta, \phi\}$ could push $\mathcal{L}(\theta, \phi)$ to $-\infty$ and yet not achieve (6). In other words, the VAE could reach the lower bound but fail to closely approximate $\mu_{gt}$. And we stress that this uniqueness issue is not a consequence of the VAE Gaussian assumptions per se; even if $q_\phi(\boldsymbol{z}|\boldsymbol{x})$ were unconstrained the same lack of uniqueness can persist.

Rather, the intrinsic difficulty is that, because the VAE model does not have access to the ground-truth low-dimensional manifold, it must implicitly rely on a density $p_\theta(\boldsymbol{x})$ defined across *all* of $\mathbb{R}^d$ as mentioned previously. Moreover, if this density converges towards infinity on the manifold during training without increasing the KL term at the same rate, the VAE cost can be unbounded from below, even in cases where (6) is not satisfied, meaning incorrect assignment of probability mass.

To conclude, the key take-home message from this section is that, at least in principle, VAE Gaussian assumptions need not actually be the root cause of any failure to recover ground-truth distributions. Instead we expose a structural deficiency that lies elsewhere, namely, the non-uniqueness of solutions that can optimize the VAE objective without necessarily learning a close approximation to $\mu_{gt}$. But to probe this issue further and motivate possible workarounds, it is critical to further disambiguate these optimal solutions and their relationship with ground-truth manifolds. This will be the task of Section 3, where we will explicitly differentiate the problem of locating the correct ground-truth manifold, from the task of learning the correct probability measure *within* the manifold.

Note that the only comparable prior work we are aware of related to the results in this section comes from Doersch (2016), where the implications of adopting Gaussian encoder/decoder pairs in the specialized case of $r = d = 1$ are briefly considered. Moreover, the analysis there requires additional much stronger assumptions than ours, namely, that $p_{gt}(\boldsymbol{x})$ should be nonzero and infinitely differentiable everywhere in the requisite 1D ambient space. These requirements of course exclude essentially all practical usage regimes where $d = r > 1$ or $d > r$, or when ground-truth densities are not sufficiently smooth.

---

[2]Note that (6) is only framed in this technical way to accommodate the difficulty of comparing a measure $\mu_{gt}$ restricted to $\boldsymbol{\chi}$ with the VAE density $p_\theta(\boldsymbol{x})$ defined everywhere in $\mathbb{R}^d$. See (Dai & Wipf, 2019) for details.

## 3 OPTIMAL SOLUTIONS AND THE GROUND TRUTH MANIFOLD

We will now more closely examine the properties of optimal $\kappa$-simple VAE solutions, and in particular, the degree to which we might expect them to at least reflect the true $\chi$, even if perhaps not the correct probability measure $\mu_{gt}$ defined within $\chi$. To do so, we must first consider some *necessary* conditions for VAE optima:

**Theorem 3** *Let $\{\theta_\gamma^*, \phi_\gamma^*\}$ denote an optimal $\kappa$-simple VAE solution (with $\kappa \geq r$) where the decoder variance $\gamma$ is fixed (i.e., it is the sole unoptimized parameter). Moreover, we assume that $\mu_{gt}$ is not a Gaussian distribution when $d = r$.[3] Then for any $\gamma > 0$, there exists a $\gamma' < \gamma$ such that $\mathcal{L}(\theta_{\gamma'}^*, \phi_{\gamma'}^*) < \mathcal{L}(\theta_\gamma^*, \phi_\gamma^*)$.*

This result implies that we can always reduce the VAE cost by choosing a smaller value of $\gamma$, and hence, if $\gamma$ is not constrained, it must be that $\gamma \to 0$ if we wish to minimize (2). Despite this necessary optimality condition, in existing practical VAE applications, it is standard to fix $\gamma \approx 1$ during training. This is equivalent to simply adopting a non-adaptive squared-error loss for the decoder and, at least in part, likely contributes to unrealistic/blurry VAE-generated samples (Bousquet et al., 2017). Regardless, there are more significant consequences of this intrinsic favoritism for $\gamma \to 0$, in particular as related to reconstructing data drawn from the ground-truth manifold $\chi$:

**Theorem 4** *Applying the same conditions and definitions as in Theorem 3, then for all $\boldsymbol{x}$ drawn from $\mu_{gt}$, we also have that*

$$\lim_{\gamma \to 0} f_{\mu_x} \left[ f_{\mu_z}(\boldsymbol{x}; \phi_\gamma^*) + f_{S_z}(\boldsymbol{x}; \phi_\gamma^*)\boldsymbol{\varepsilon}; \theta_\gamma^* \right] = \lim_{\gamma \to 0} f_{\mu_x} \left[ f_{\mu_z}(\boldsymbol{x}; \phi_\gamma^*); \theta_\gamma^* \right] = \boldsymbol{x}, \quad \forall \boldsymbol{\varepsilon} \in \mathbb{R}^\kappa. \quad (7)$$

By design any random draw $\boldsymbol{z} \sim q_{\phi_\gamma^*}(\boldsymbol{z}|\boldsymbol{x})$ can be expressed as $f_{\mu_z}(\boldsymbol{x}; \phi_\gamma^*) + f_{S_z}(\boldsymbol{x}; \phi_\gamma^*)\boldsymbol{\varepsilon}$ for some $\boldsymbol{\varepsilon} \sim \mathcal{N}(\boldsymbol{\varepsilon}|\boldsymbol{0}, \boldsymbol{I})$. From this vantage point then, (7) effectively indicates that any $\boldsymbol{x} \in \chi$ will be perfectly reconstructed by the VAE encoder/decoder pair at globally optimal solutions, achieving this necessary condition despite any possible stochastic corrupting factor $f_{S_z}(\boldsymbol{x}; \phi_\gamma^*)\boldsymbol{\varepsilon}$.

But still further insights can be obtained when we more closely inspect the VAE objective function behavior at arbitrarily small but explicitly nonzero values of $\gamma$. In particular, when $\kappa = r$ (meaning $\boldsymbol{z}$ has no superfluous capacity), Theorem 4 and attendant analyses in (Dai & Wipf, 2019) ultimately imply that the squared eigenvalues of $f_{S_z}(\boldsymbol{x}; \phi_\gamma^*)$ will become arbitrarily small at a rate proportional to $\gamma$, meaning $\frac{1}{\sqrt{\gamma}} f_{S_z}(\boldsymbol{x}; \phi_\gamma^*) \approx O(1)$ under mild conditions. It then follows that the VAE data term integrand from (2), in the neighborhood around optimal solutions, behaves as $-2\mathbb{E}_{q_{\phi_\gamma^*}(\boldsymbol{z}|\boldsymbol{x})} \left[ \log p_{\theta_\gamma^*}(\boldsymbol{x}|\boldsymbol{z}) \right] =$

$$2\mathbb{E}_{q_{\phi_\gamma^*}(\boldsymbol{z}|\boldsymbol{x})} \left[ \frac{1}{\gamma} \left\| \boldsymbol{x} - f_{\mu_x} \left[ \boldsymbol{z}; \theta_\gamma^* \right] \right\|_2^2 \right] + d \log 2\pi\gamma \approx \mathbb{E}_{q_{\phi_\gamma^*}(\boldsymbol{z}|\boldsymbol{x})} [O(1)] + d \log 2\pi\gamma = d \log \gamma + O(1). \quad (8)$$

This expression can be derived by excluding the higher-order terms of a Taylor series approximation of $f_{\mu_x} \left[ f_{\mu_z}(\boldsymbol{x}; \phi_\gamma^*) + f_{S_z}(\boldsymbol{x}; \phi_\gamma^*)\boldsymbol{\varepsilon}; \theta_\gamma^* \right]$ around the point $f_{\mu_z}(\boldsymbol{x}; \phi_\gamma^*)$, which will be relatively tight under the stated conditions. But because $2\mathbb{E}_{q_{\phi_\gamma^*}(\boldsymbol{z}|\boldsymbol{x})} \left[ \frac{1}{\gamma} \left\| \boldsymbol{x} - f_{\mu_x} \left[ \boldsymbol{z}; \theta_\gamma^* \right] \right\|_2^2 \right] \geq 0$, a theoretical lower bound on (8) is given by $d \log 2\pi\gamma \equiv d \log \gamma + O(1)$. So in this sense (8) cannot be significantly lowered further.

This observation is significant when we consider the inclusion of addition latent dimensions by allowing $\kappa > r$. Clearly based on the analysis above, adding dimensions to $\boldsymbol{z}$ *cannot* improve the value of the VAE data term in any meaningful way. However, it can have a detrimental impact on the the KL regularization factor in the $\gamma \to 0$ regime, where

$$2\mathbb{KL}\left[q_\phi(\boldsymbol{z}|\boldsymbol{x})||p(\boldsymbol{z})\right] \equiv \text{trace} \left[\boldsymbol{\Sigma}_z\right] + \|\boldsymbol{\mu}_z\|_2^2 - \log |\boldsymbol{\Sigma}_z| \approx -\hat{r} \log \gamma + O(1). \quad (9)$$

Here $\hat{r}$ denotes the number of eigenvalues $\{\lambda_j(\gamma)\}_{j=1}^\kappa$ of $f_{S_z}(\boldsymbol{x}; \phi_\gamma^*)$ (or equivalently $\boldsymbol{\Sigma}_z$) that satisfy $\lambda_j(\gamma) \to 0$ if $\gamma \to 0$. $\hat{r}$ can be viewed as an estimate of how many low-noise latent dimensions

---

[3]This requirement is only included to avoid a practically irrelevant form of non-uniqueness that exists with full, non-degenerate Gaussian distributions.

the VAE model is preserving to reconstruct $\boldsymbol{x}$. Based on (9), there is obvious pressure to make $\hat{r}$ as small as possible, at least without disrupting the data fit. The smallest possible value is $\hat{r} = r$, since it is not difficult to show that any value below this will contribute consequential reconstruction errors, causing $2\mathbb{E}_{q_{\phi_\gamma^*}(\boldsymbol{z}|\boldsymbol{x})}\left[\frac{1}{\gamma}\left\|\boldsymbol{x} - f_{\mu_x}\left[\boldsymbol{z}; \theta_\gamma^*\right]\right\|_2^2\right]$ to grow at a rate of $\Omega\left(\frac{1}{\gamma}\right)$, pushing the entire cost function towards infinity.[4]

Therefore, *in the neighborhood of optimal solutions the VAE will naturally seek to produce perfect reconstructions using the fewest number of clean, low-noise latent dimensions*, meaning dimensions whereby $q_\phi(\boldsymbol{z}|\boldsymbol{x})$ has negligible variance. For superfluous dimensions that are unnecessary for representing $\boldsymbol{x}$, the associated encoder variance in these directions can be pushed to one. This will optimize $\mathbb{KL}\left[q_\phi(\boldsymbol{z}|\boldsymbol{x})||p(\boldsymbol{z})\right]$ along these directions, and the decoder can selectively block the residual randomness to avoid influencing the reconstructions per Theorem 4. So in this sense the VAE is capable of learning a minimal representation of the ground-truth manifold $\boldsymbol{\chi}$ when $r < \kappa$.

But we must emphasize that the VAE can learn $\boldsymbol{\chi}$ independently of the actual distribution $\mu_{gt}$ within $\boldsymbol{\chi}$. Addressing the latter is a completely separate issue from achieving the perfect reconstruction error defined by Theorem 4. This fact can be understood within the context of a traditional PCA-like model, which is perfectly capable of learning a low-dimensional subspace containing some training data without actually learning the distribution of the data within this subspace. The central issue is that there exists an intrinsic bias associated with the VAE objective such that *fitting the distribution within the manifold will be completely neglected whenever there exists the chance for even an infinitesimally better approximation of the manifold itself.*

Stated differently, if VAE model parameters have learned a near optimal, parsimonious latent mapping onto $\boldsymbol{\chi}$ using $\gamma \approx 0$, then the VAE cost will scale as $(d - r) \log \gamma$ regardless of $\mu_{gt}$. Hence there remains a *huge* incentive to reduce the reconstruction error still further, allowing $\gamma$ to push even closer to zero and the cost closer to $-\infty$. And if we constrain $\gamma$ to be sufficiently large so as to prevent this from happening, then we risk degrading/blurring the reconstructions and widening the gap between $q_\phi(\boldsymbol{z}|\boldsymbol{x})$ and $p_\theta(\boldsymbol{z}|\boldsymbol{x})$, which can also compromise estimation of $\mu_{gt}$. Fortunately though, as will be discussed next there is a convenient way around this dilemma by exploiting the fact that this dominating $(d - r) \log \gamma$ factor goes away when $d = r$.

## 4 FROM THEORY TO PRACTICAL VAE ENHANCEMENTS

Sections 2 and 3 have exposed a collection of VAE properties with useful diagnostic value in and of themselves. But the practical utility of these results, beyond the underappreciated benefit of learning $\gamma$, warrant further exploration. In this regard, suppose we wish to develop a generative model of high-dimensional data $\boldsymbol{x} \in \boldsymbol{\chi}$ where unknown low-dimensional structure is significant (i.e., the $r < d$ case with $r$ unknown). The results from Section 3 indicate that the VAE can partially handle this situation by learning a parsimonious representation of low-dimensional manifolds, but not necessarily the correct probability measure $\mu_{gt}$ within such a manifold. In quantitative terms, this means that a decoder $p_\theta(\boldsymbol{x}|\boldsymbol{z})$ will map all samples from an encoder $q_\phi(\boldsymbol{z}|\boldsymbol{x})$ to the correct manifold such that the reconstruction error is negligible for any $\boldsymbol{x} \in \boldsymbol{\chi}$. But if the measure $\mu_{gt}$ on $\boldsymbol{\chi}$ has not been accurately estimated, then

$$q_\phi(\boldsymbol{z}) \triangleq \int_{\boldsymbol{\chi}} q_\phi(\boldsymbol{z}|\boldsymbol{x})\mu_{gt}(d\boldsymbol{x}) \not\approx \int_{\mathbb{R}^d} p_\theta(\boldsymbol{z}|\boldsymbol{x})p_\theta(\boldsymbol{x})d\boldsymbol{x} = \int_{\mathbb{R}^d} p_\theta(\boldsymbol{x}|\boldsymbol{z})p(\boldsymbol{z})d\boldsymbol{x} = \mathcal{N}(\boldsymbol{z}|\boldsymbol{0}, \boldsymbol{I}), \quad (10)$$

where $q_\phi(\boldsymbol{z})$ is sometimes referred to as the aggregated posterior (Makhzani et al., 2016). In other words, the distribution of the latent samples drawn from the encoder distribution, when averaged across the training data, will have lingering latent structure that is errantly incongruous with the original isotropic Gaussian prior. This then disrupts the pivotal ancestral sampling capability of the VAE, implying that samples drawn from $\mathcal{N}(\boldsymbol{z}|0, I)$ and then passed through the decoder $p_\theta(\boldsymbol{x}|\boldsymbol{z})$ will *not* closely approximate $\mu_{gt}$. Fortunately, our analysis suggests the following two-stage remedy:

1. Given $n$ observed samples $\{\boldsymbol{x}^{(i)}\}_{i=1}^n$, train a $\kappa$-simple VAE, with $\kappa \geq r$, to estimate the unknown $r$-dimensional ground-truth manifold $\boldsymbol{\chi}$ embedded in $\mathbb{R}^d$ using a minimal number of active latent dimensions. Generate latent samples $\{\boldsymbol{z}^{(i)}\}_{i=1}^n$ via $\boldsymbol{z}^{(i)} \sim q_\phi(\boldsymbol{z}|\boldsymbol{x}^{(i)})$. By design, these samples will be distributed as $q_\phi(\boldsymbol{z})$, but likely not $\mathcal{N}(\boldsymbol{z}|\boldsymbol{0}, \boldsymbol{I})$.

---

[4]Note that $\inf_{\gamma>0} \frac{C}{\gamma} + \log \gamma = \infty$ for any $C > 0$.

2. Train a second $\kappa$-simple VAE, with independent parameters $\{\theta', \phi'\}$ and latent representation $\boldsymbol{u}$, to learn the unknown distribution $q_\phi(\boldsymbol{z})$, i.e., treat $q_\phi(\boldsymbol{z})$ as a new ground-truth distribution and use samples $\{\boldsymbol{z}^{(i)}\}_{i=1}^{n}$ to learn it.

3. Samples approximating the *original* ground-truth $\mu_{gt}$ can then be formed via the extended ancestral process $\boldsymbol{u} \sim \mathcal{N}(\boldsymbol{u}|\boldsymbol{0}, \boldsymbol{I})$, $\boldsymbol{z} \sim p_{\theta'}(\boldsymbol{z}|\boldsymbol{u})$, and finally $\boldsymbol{x} \sim p_\theta(\boldsymbol{x}|\boldsymbol{z})$.

The efficacy of the second-stage VAE from above is based on the following. If the first stage was successful, then even though they will not generally resemble $\mathcal{N}(\boldsymbol{z}|\boldsymbol{0}, \boldsymbol{I})$, samples from $q_\phi(\boldsymbol{z})$ will nonetheless have nonzero measure across the full ambient space $\mathbb{R}^\kappa$. If $\kappa = r$, this occurs because the entire latent space is needed to represent an $r$-dimensional manifold, and if $\kappa > r$, then the extra latent dimensions will be naturally filled in via randomness introduced along dimensions associated with nonzero eigenvalues of the decoder covariance $\boldsymbol{\Sigma}_z$ per the analysis in Section 3.

Consequently, as long as we set $\kappa \geq r$, *the operational regime of the second-stage VAE is effectively equivalent to the situation described in Section 2.1 where the manifold dimension is equal to the ambient dimension*.[5] And as we have already shown there via Theorem 1, the VAE can readily handle this situation, since in the narrow context of the second-stage VAE, $d = r = \kappa$, the troublesome $(d - r) \log \gamma$ factor becomes zero, and any globally minimizing solution is uniquely matched to the new ground-truth distribution $q_\phi(\boldsymbol{z})$. Consequently, the revised aggregated posterior $q_{\phi'}(\boldsymbol{u})$ produced by the second-stage VAE should now closely resemble $\mathcal{N}(\boldsymbol{u}|\boldsymbol{0}, \boldsymbol{I})$. And importantly, because we generally assume that $d \gg \kappa \geq r$, we have found that the second-stage VAE can be quite small.

It should also be emphasized that concatenating the two VAE stages and jointly training does *not* generally improve the performance. If trained jointly the few extra second-stage parameters can simply be hijacked by the dominant influence of the first stage reconstruction term and forced to work on an incrementally better fit of the manifold rather than addressing the critical mismatch between $q_\phi(\boldsymbol{z})$ and $\mathcal{N}(\boldsymbol{u}|\boldsymbol{0}, \boldsymbol{I})$. This observation can be empirically tested, which we have done in multiple ways. For example, we have tried fusing the respective encoders and decoders from the first and second stages to train what amounts to a slightly more complex single VAE model. We have also tried merging the two stages including the associated penalty terms. In both cases, joint training does not help at all as expected, with average performance no better than the first stage VAE (which contains the vast majority of parameters). Consequently, although perhaps counterintuitive, separate training of these two VAE stages is actually critical to achieving high quality results as will be demonstrated next.

## 5 Quantitative Comparisons of Generated Sample Quality

We first present quantitative evaluation of novel generated samples using the large-scale testing protocol of GAN models from (Lucic et al., 2018). In this regard, GANs are well-known to dramatically outperform existing VAE approaches in terms of the Fréchet Inception Distance (FID) score (Heusel et al., 2017) and related quantitative metrics. For fair comparison, (Lucic et al., 2018) adopted a common neutral architecture for all models, with generator and discriminator networks based on InfoGAN (Chen et al., 2016a); the point here is standardized comparisons, not tuning arbitrarily-large networks to achieve the lowest possible absolute FID values. We applied the same architecture to our first-stage VAE decoder and encoder networks respectively for direct comparison. For the low-dimensional second-stage VAE we used small, 3-layer networks contributing negligible additional parameters beyond the first stage (see (Dai & Wipf, 2019) for further design details).

We evaluated our proposed VAE pipeline, henceforth denoted as *2-Stage VAE*, against three baseline VAE models differing only in the decoder output layer: a Gaussian layer with fixed $\gamma$, a Gaussian layer with a learned $\gamma$, and a cross-entropy layer as has been adopted in several previous applications involving images (Chen et al., 2016b). We also tested the Gaussian decoder VAE model (with learned $\gamma$) combined with an encoder augmented with normalizing flows (Rezende & Mohamed, 2015), as well as the recently proposed Wasserstein autoencoder (WAE) (Tolstikhin et al., 2018) which maintains a VAE-like structure. All of these models were adapted to use the same neutral architecture from (Lucic et al., 2018). Note also that the WAE includes two variants, referred to

---

[5]Note that if a regular autoencoder were used to replace the first-stage VAE, then this would no longer be the case, so indeed a VAE is required for both stages unless we have strong prior knowledge such that we may confidently set $\kappa \approx r$.

as WAE-MMD and WAE-GAN because different Maximum Mean Discrepancy (MMD) and GAN regularization factors are involved. We conduct experiments using the former because it does not involve potentially-unstable adversarial training, consistent with the other VAE baselines.[6] Additionally, we present results from (Lucic et al., 2018) involving numerous competing GAN models, including MM GAN (Goodfellow et al., 2014), WGAN (Arjovsky et al., 2017), WGAN-GP (Gulra-jani et al., 2017), NS GAN (Fedus et al., 2017), DRAGAN (Kodali et al., 2017), LS GAN (Mao et al., 2017) and BEGAN (Berthelot et al., 2017). Testing is conducted across four significantly different datasets: MNIST (LeCun et al., 1998), Fashion MNIST (Xiao et al., 2017), CIFAR-10 (Krizhevsky & Hinton, 2009) and CelebA (Liu et al., 2015).

For each dataset we executed 10 independent trials and report the mean and standard deviation of the FID scores in Table 1.[7] No effort was made to tune VAE training hyperparameters (e.g., learning rates, etc.); rather a single generic setting was first agnostically selected and then applied to all VAE-like models (including the WAE-MMD). As an analogous baseline, we also report the value of the best GAN model for each dataset when trained using suggested settings from the authors; no single model was optimal across all datasets, so these values represent the best performance from different, dataset-dependent GANs. Even so, our single 2-Stage VAE is still better on two of four datasets, and in aggregate, better than any individual GAN model. For example, when averaged across datasets, the mean FID score for any individual GAN trained with suggested settings was always approximately 45 or higher (see (Lucic et al., 2018)[Figure 4]), while our analogous 2-Stage VAE maintained a mean below 40. The other VAE baselines were not competitive. Note also that the relatively poor performance of the WAE-MMD on MNIST and Fashion MNIST data can be attributed to the sensitivity of this approach to the value of $\kappa$, which for consistency with other models was fixed at $\kappa = 64$ for all experiments. This value is likely much larger than actually needed for these simpler data types (meaning $r \ll 64$), and the WAE-MMD model can potentially be more reliant on having some $\kappa \approx r$. For head-to-head empirical tests of robustness to $\kappa$, please see (Dai & Wipf, 2019).

Table 1 also displays FID scores from GAN models evaluated using hyperparameters obtained from a large-scale search executed independently across each dataset to achieve the best results; 100 settings per model per dataset, plus an optimal, data-dependent stopping criteria as described in (Lucic et al., 2018). Within this broader paradigm, cases of severe mode collapse were omitted when computing final GAN FID averages. Despite these considerable GAN-specific advantages, the FID performance of the default 2-Stage VAE is well within the range of the heavily-optimized GAN models for each dataset unlike the other VAE baselines. Overall then, these results represent the first demonstration of a VAE pipeline capable of competing with GANs in the arena of generated sample quality. Additionally, representative samples produced using our 2-Stage VAE model can be found in (Dai & Wipf, 2019).

Beyond the neutral testing platform from (Lucic et al., 2018), we also consider additional comparisons using the architecture and experimental setup from (Tolstikhin et al., 2018) explicitly designed for applying WAE models to CelebA data. In particular, we adopt the exact same encoder-decoder networks as the WAE models, and train using the same number of epochs. We do not tune any hyperparameters whatsoever, and apply the same small second-stage VAE as used in previous experiments. As before, the second-stage size is a small fraction of the first stage, so any benefit is not simply the consequence of a larger network structure. Results are reported in Table 2, where the 2-Stage VAE even outperforms the WAE-GAN model, which has the advantage of adversarial training tuned for this combination of data and network architecture.

## 6  EXPERIMENTAL CORROBORATION OF THEORETICAL RESULTS

The true test of any theoretical contribution is the degree to which it leads to useful, empirically-testable predictions about behavior in real-world settings. In the present context, although our theory from Sections 2 and 3 involves some unavoidable simplifying assumptions, it nonetheless makes predictions that can be tested under practically-relevant conditions where these assumptions may

---

[6]Later we compare against both WAE-MMD and WAE-GAN using the setup from (Tolstikhin et al., 2018).

[7]All reported FID scores for VAE and GAN models were computed using TensorFlow (https://github.com/bioinf-jku/TTUR). We have found that alternative PyTorch implementations (https://github.com/mseitzer/pytorch-fid) can produce different values in some circumstances.

|  |  | MNIST | Fashion | CIFAR-10 | CelebA |
|---|---|---|---|---|---|
| optimized, data-dependent settings | MM GAN | $9.8 \pm 0.9$ | $29.6 \pm 1.6$ | $72.7 \pm 3.6$ | $65.6 \pm 4.2$ |
|  | NS GAN | $6.8 \pm 0.5$ | $26.5 \pm 1.6$ | $58.5 \pm 1.9$ | $55.0 \pm 3.3$ |
|  | LSGAN | $7.8 \pm 0.6$ | $30.7 \pm 2.2$ | $87.1 \pm 47.5$ | $53.9 \pm 2.8$ |
|  | WGAN | $6.7 \pm 0.4$ | $21.5 \pm 1.6$ | $55.2 \pm 2.3$ | $41.3 \pm 2.0$ |
|  | WGAN GP | $20.3 \pm 5.0$ | $24.5 \pm 2.1$ | $55.8 \pm 0.9$ | $30.3 \pm 1.0$ |
|  | DRAGAN | $7.6 \pm 0.4$ | $27.7 \pm 1.2$ | $69.8 \pm 2.0$ | $42.3 \pm 3.0$ |
|  | BEGAN | $13.1 \pm 1.0$ | $22.9 \pm 0.9$ | $71.4 \pm 1.6$ | $38.9 \pm 0.9$ |
| default settings | Best GAN | $\sim 10$ | $\sim 32$ | $\sim 70$ | $\sim 49$ |
|  | VAE (cross-entr.) | $16.6 \pm 0.4$ | $43.6 \pm 0.7$ | $106.0 \pm 1.0$ | $53.3 \pm 0.6$ |
|  | VAE (fixed $\gamma$) | $52.0 \pm 0.6$ | $84.6 \pm 0.9$ | $160.5 \pm 1.1$ | $55.9 \pm 0.6$ |
|  | VAE (learned $\gamma$) | $54.5 \pm 1.0$ | $60.0 \pm 1.1$ | $76.7 \pm 0.8$ | $60.5 \pm 0.6$ |
|  | VAE + Flow | $54.8 \pm 2.8$ | $62.1 \pm 1.6$ | $81.2 \pm 2.0$ | $65.7 \pm 2.8$ |
|  | WAE-MMD | $115.0 \pm 1.1$ | $101.7 \pm 0.8$ | $80.9 \pm 0.4$ | $62.9 \pm 0.8$ |
|  | 2-Stage VAE (ours) | $12.6 \pm 1.5$ | $29.3 \pm 1.0$ | $72.9 \pm 0.9$ | $44.4 \pm 0.7$ |

Table 1: FID score comparisons using neutral architecture. For all GAN-based models listed in the top section of the table, reported values represent the optimal FID obtained across a large-scale hyperparameter search conducted separately for each dataset (Lucic et al., 2018). Outlier cases (e.g., severe mode collapse) were omitted, which would have otherwise increased these GAN FID scores. In the lower section of the table, the label *Best GAN* indicates the lowest FID produced across all GAN approaches for each dataset when trained using settings suggested by original authors; these approximate values were extracted from (Lucic et al., 2018)[Figure 4]. For the VAE results (including WAE), only a single default setting was adopted across all datasets and models (no tuning whatsoever), and no cases of mode collapse were removed. Note that specialized architectures and/or random seed optimization can potentially improve the FID for all models reported here.

|  | VAE | WAE-MMD | WAE-GAN | 2-Stage VAE (ours) |
|---|---|---|---|---|
| CelebA FID | 63 | 55 | 42 | 34 |

Table 2: FID scores on CelebA data obtained using the network structure and training protocol from (Tolstikhin et al., 2018). For the 2-Stage VAE, we apply the exact same architecture and training epochs without any tuning of hyperparameters.

not strictly hold. We now present the results of such tests, which provide strong confirmation of our previous analysis. In particular, after providing validation of Theorems 3 and 4, we explicitly demonstrate that the second stage of our 2-Stage VAE model can reduce the gap between $q(\boldsymbol{z})$ and $p(\boldsymbol{z})$.

**Validation of Theorem 3:** This theorem implies that $\gamma$ will converge to zero at any global minimum of the stated VAE objective under consideration. Figure 1a presents empirical support for this result, where indeed the decoder variance $\gamma$ does tend towards zero during training (red line). This then allows for tighter image reconstructions (dark blue curve) with lower average squared error, i.e., a better manifold fit as expected.

**Validation of Theorem 4:** Figure 1b bolsters this theorem, and the attendant analysis which follows in Section 3, by showcasing the dissimilar impact of noise factors applied to different directions in the latent space before passage through the decoder mean network $f_{\mu_x}$. In a direction where an eigenvalue $\lambda_j$ of $\boldsymbol{\Sigma}_z$ is large (i.e., a superfluous dimension), a random perturbation is completely muted by the decoder as predicted. In contrast, in directions where such eigenvalues are small (i.e., needed for representing the manifold), varying the input causes large changes in the image space reflecting reasonable movement along the correct manifold.

**Reduced Mismatch between $q_\phi(\boldsymbol{z})$ and $p(\boldsymbol{z})$:** Although the VAE with a learnable $\gamma$ can achieve high-quality reconstructions, the associated aggregated posterior is still likely not close to a standard Gaussian distribution as implied by (10). This mismatch then disrupts the critical ancestral sampling

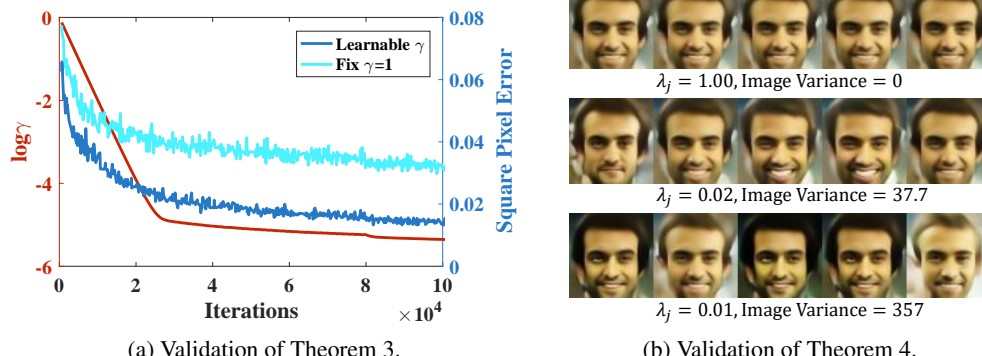

(a) Validation of Theorem 3.    (b) Validation of Theorem 4.

Figure 1: (*a*) The red line shows the evolution of $\log \gamma$, converging close to 0 during training as expected. The two blue curves compare the associated pixel-wise reconstruction errors with $\gamma$ fixed at 1 and with a learnable $\gamma$ respectively. (*b*) The $j$-th eigenvalue of $\Sigma_z$, denoted $\lambda_j$, should be very close to either 0 or 1 as argued in Section 3. When $\lambda_j$ is close to 0, injecting noise along the corresponding direction will cause a large variance in the reconstructed image, meaning this direction is an informative one needed for representing the manifold. In contrast, if $\lambda_j$ is close to 1, the addition of noise does not make any appreciable difference in the reconstructed image, indicating that the corresponding dimension is a superfluous one that has been ignored/blocked by the decoder.

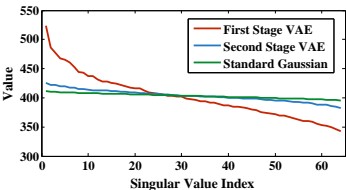

Figure 2: Singular value spectrums of latent sample matrices drawn from $q_\phi(z)$ (first stage) and $q_{\phi'}(u)$ (enhanced second stage).

Table 3: Maximum mean discrepancy between $\mathcal{N}(0, I)$ and $q_\phi(z)$ (first stage); likewise for $q_{\phi'}(u)$ (second stage).

|  | First Stage | Second Stage |
|---|---|---|
| MNIST | 2.85 | 0.43 |
| Fashion | 1.37 | 0.40 |
| Cifar10 | 1.08 | 0.00 |
| CelabA | 7.42 | 0.29 |

process. As we have previously argued, the proposed 2-Stage VAE has the ability to overcome this issue and achieve a standard Gaussian aggregated posterior, or at least nearly so. As empirical evidence for this claim, Figure 2 displays the singular value spectrum of latent sample matrices $Z = \{z^{(i)}\}_{i=1}^n$ drawn from $q_\phi(z)$ (first stage), and $U = \{u^{(i)}\}_{i=1}^n$ drawn from $q_{\phi'}(u)$ (enhanced second stage). As expected, the latter is much closer to the spectrum from an analogous i.i.d. $\mathcal{N}(0, I)$ matrix. We also used these same sample matrices to estimate the MMD metric (Gretton et al., 2007) between $\mathcal{N}(0, I)$ and the aggregated posterior distributions from the first and second stages in Table 3. Clearly the second stage has dramatically reduced the difference from $\mathcal{N}(0, I)$ as quantified by the MMD. Overall, these results indicate a superior latent representation, providing high-level support for our 2-Stage VAE proposal.

# 7  DISCUSSION

It is often assumed that there exists an unavoidable trade-off between the stable training, valuable attendant encoder network, and resistance to mode collapse of VAEs, versus the impressive visual quality of images produced by GANs. While we certainly are not claiming that our two-stage VAE model is superior to the latest and greatest GAN-based architecture in terms of the realism of generated samples, we do strongly believe that this work at least narrows that gap substantially such that VAEs are worth considering in a broader range of applications. For further results and discussion, including consideration of broader VAE modeling paradigms and the identifiability of disentangled representations, please see (Dai & Wipf, 2019).

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
