# OpenReview forum: "Diagnosing and Enhancing VAE Models"
_ICLR.cc/2019/Conference_

### Official Review · AnonReviewer1 · 2018-11-01
**Careful analysis of Gaussian VAEs yields valuable insights and training procedure**

**Rating:** 7
**Confidence:** 4

**Review:**

Overview:
I thank the authors for their interesting and detailed work in this paper. I believe it has the potential to provide strong value to the community interested in using VAEs with an explicit and simple parameterization of the approximate posterior and likelihood as Gaussian. Gaussianity can be appropriate in many cases where no sequential or discrete structure needs to be induced in the model. I find the mathematical arguments interesting and enlightening. However, the authors somewhat mischaracterize the scope of applicability of VAE models in contemporary machine learning, and don't show familiarity with the broad literature around VAEs outside of this case (that is, where a Gaussian model of the output would be manifestly inappropriate). Since the core of the paper is valuable and salvageable from a clarity standpoint, my comments below are geared towards what changes the authors may make to move this paper into the "pass" category.

Pros:
- Mathematical insights are well reasoned and interesting. Based on the insight from the analysis in the supplementary materials, the authors propose a two-stage VAE which separate learning the a parsimonious representation of the low-dimensional (lower than the ambient dimension of the input space), and the training a second VAE to learn the unknown approximate posterior. The two-stage training procedure is both theoretically motivated and appears to enhance the output quality of VAEs w.r.t. FID score, making them rival GAN architectures on this metric.

Cons:
- The title and general tone of the paper is too broad: it is only VAE models with Gaussian approximate posteriors and likelihoods. This is hardly the norm for most applications, contrary to the claims of the authors. VAEs are commonly used for discrete random variables, for example. Many cases where VAEs are applied cannot use a Gaussian assumption for the likelihood, which is the key requirement for the proofs in the supplement to be valid (then, the true posterior is also Gaussian, and the KL divergence between that and the approximate posterior can be driven to zero during optimization--clearly a Gaussian approximate posterior will never have zero KL divergence with a non-Gaussian true posterior).
- None of the proofs consider the approximation error garnered by only having access to empirical samples through a sample of the ground truth population. (The ground-truth distribution must be defined with respect to the population rather just the dataset in hand, otherwise we lose all generalizability from a model.) Moreover, the proofs hold asymptotically. Generalization bounds and error from finite time approximations are very pertinent issues and these are ignored by the presented analyses. Such concerns have motivated many of the recent developments in approximate posterior distributions. Overall, the paper contains little evidence of familiarity with the recent advances in approximate Bayesian inference that have occurred over the past two years.
- A central claim of the paper is that the two-stage VAE obviates the need for highly adaptive approximate posteriors. However, no comparison against those models is done in the paper. How does a two-stage VAE compare against one with, e.g., a normalizing flow approximate posterior? I acknowledge that the purpose of the paper was to argue for the Gaussianity assumption as less stringent than previously believed, but all of the mathematical arguments take place in an imagined world with infinite time and unbounded access to the population distribution. This is not really the domain of interest in modern computational statistics / machine learning, where issues of generalization and computational efficiency are paramount.
- While the mathematical insights are well developed, the specifics of the algorithm used to implement the two-stage VAE are a little opaque. Ancestral sampling now takes place using latent samples from a second VAE. An algorithm box is badly needed for reproducibility.

Recommendations / Typos

I noted a few typos and omissions that need correction.

- Generally, the mathematical proofs in section 7 of the supplement are clear. At the top of page 11, though, the paragraph correctly begins by stating that the composition of invertible functions is invertible, but fails to establish that G is also invertible. Clearly it is so by construction, but the explicit reasons should be stated (as a prior sentence promises), and so I assume this is an accidental omission.
- The title of Section 8.1 has a typo: clearly is it is the negative log of p_{theta_t} (x) which approaches its infimum rather than p_{theta_t} (x) approaching negative infinity.
- Equation (4): the true posterior has an x as its argument instead of the latent z.
- Missing parenthesis under Case 2 and wrong indentation. This analysis also seems to be cut off. Is the case r > d relevant here?

* EDIT: I have read the authors' detailed response. It has clarified a few key issues, and convinced me of the value to the community for publication in its present (slightly edited according to the reviwers' feedback) form. I would like to see this published and discussed at ICLR and have revised my score accordingly. *

---

> ### Author Response · Authors · 2018-11-10
> **Response to AnonReviewer1 -- Part 3**
>
> - Reviewer Comment:  No comparisons against VAE models with more flexible approximate posteriors such as those produced via normalizing flows
>
> Our Response:  We agree that more flexible, explicitly non-Gaussian approximate posteriors have recently been proposed, such as the many flavors that utilize normalizing flows.  But such models have not as of yet been objectively shown to improve sampling quality (see comments above) despite the tremendous community-wide incentive to publish such a demonstration.  Moreover, the added flexibility often comes with a significant cost (e.g., increased training difficulty, more expensive inference).  Furthermore, if we consider broader VAE modifications beyond just the encoder, then even within this wider domain, the only VAE-related enhancement we are aware of that objectively/quantitatively produces improved samples is the WAE model from ICLR this year (Tolstikhin et al., 2018), which is already explicitly addressed in Section 5 of our submission.  Consequently, unless there is some very recent reference we may have missed, our experiments represent the state-of-the-art for non-adversarial VAE/autoencoer-based structures in term of the objective evaluation of generated samples, and the first to close the gap with GANs (this is also consistent with the comments from AnonReviewer2).
>
>
> - Reviewer Comment:  Some details about the proposed 2-stage process are unclear
>
> Although there was unfortunately no space for a separate algorithm box in our submission, the three bullet points on page 6 describe the specific process we used.  Note that the ancestral sampling required is very straightforward as described in bullet point 3 on page 6.  This is exactly what we followed for generating new samples via our method, but we are happy to provide further clarification if the reviewer has a specific suggestion.
>
>
> - Reviewer Comment:  Recommendations/Typos
>
> Our Response:  We sincerely appreciate the effort in finding typos and checking the proofs.  We have corrected each of the cases the reviewer uncovered.  This will certainly be of benefit to future readers.  Additionally, r can never be greater than d, because r is the manifold dimension within the ambient space of dimension d.

---

> ### Author Response · Authors · 2018-11-10
> **Response to AnonReviewer1 -- Part 2**
>
> - Reviewer Comment:  Approximation error arising from finite samples not addressed; missing references to advances in approximate inference
>
> Our Response:  In an ideal world we would obviously like to have optimal finite sample approximations that closely reflect practical testing scenarios.  But such a bar is impossibly high at this point.  Overall, we believe the value of theoretical inquiry into asymptotic regimes (i.e., population data rather than finite samples) cannot be dismissed out of hand, especially when simplifying assumptions of some sort are absolutely essential in making any reasonable progress.  Even so, the true test of any theoretical contribution is the degree to which it leads to useful, empirically-testable predictions about behavior in real-world settings.  In the present context, our theory makes the seemingly counter-intuitive prediction that a simple two-stage VAE could circumvent existing problems and produce realistic samples.  We then tested this idea via the neutral DNN architecture and comprehensive experimental design from reference (Lucic et al., 2018) and it immediately worked.  It is also critical to emphasize that these experiments were designed by others to evaluate top-performing GAN models with respect to generated sample quality, they were not developed to favor our approach in any way via some carefully tuned architecture or setting.  Therefore, regardless of whether or not our theory involves asymptotic assumptions, it made testable, non-obvious predictions that were confirmed in a real-world practical environment, providing the very first VAE-based architecture that is quantitatively competitive with GANs in generating novel samples (at least with continuous data like images).  We strongly believe that this is the hallmark of a significant contribution.
>
> The reviewer also mentions that we may be unfamiliar with certain recent advances in approximate Bayesian inference, but no references were provided.  Which papers in particular is the reviewer referring to?  We are quite open to hearing about relevant work that we may have missed; however, presently we are unaware of any overlooked references that might serve to discount the message of our paper.  Note that there is an extensive recent literature developing more sophisticated VAE inference networks using normalizing flows and related. However, to the best of our knowledge, none of these works contain quantitative evaluations of generated sample quality (our focus here), and many (possibly most) do not even contain visualizations of images generated by the model.  Please see reference (van den Berg et al., "Sylvester Normalizing Flows for Variational Inference," UAI 2018) for the latest representative example we have found.  Of course our point here is not to disparage insightful papers of this type that provide significant advances in approximate inference.  Rather we are merely arguing that they seem to be somewhat out of the scope of our present submission, especially given the limited space for broader discussions.  But we can try to squeeze in more references and background perspective of this nature if the reviewer feels it could be helpful.

---

> ### Author Response · Authors · 2018-11-10
> **Response to AnonReviewer1 -- Part 1**
>
> Thanks for providing detailed comments regarding our manuscript, including constructive ideas on how to improve the presentation and clarify the context.  We address each main point in turn.
>
>
> - Reviewer Comment:  Limitation of Gaussian assumptions for likelihoods and approximate posteriors
>
> Our Response:  In the introduction, we state that the most commonly adopted distributional assumption is that the encoder and decoder are Gaussian.  This claim was based on an informal survey of numerous recent papers involving VAE models applied to continuous data (e.g., images, etc.).  However, we completely agree that VAEs can also be successfully applied to discrete data types like language models, where these Gaussian assumptions can be more problematic.  Although all of our theoretical developments are clearly framed in the context of continuous data on a manifold, we are happy to revise the introduction to better explain this issue up front.  And of course the whole point of our paper is rigorously showing that even with seemingly restrictive Gaussian assumptions, highly non-Gaussian continuous distributions can nonetheless be accurately modeled.
>
> Also, just to clarify one lingering point: although the decoder p(x|z) is defined to be Gaussian, it does not follow that the associated posterior p(z|x) will necessarily be Gaussian as well.  In fact this will usually not be the case when using deep models and parameters in general position.  However, the VAE can still push the KL divergence between p(z|x) and q(z|x) to zero even when the latter is constrained to be Gaussian as long as there exists at least some specially matched encoder-decoder parameterizations capable of pushing them together everywhere except on a space of measure zero.  This was left as an open problem under general conditions in the most highly-cited VAE tutorial (Doersch, 2016), and is what we demonstrate in Section 2.

---

### Official Review · AnonReviewer3 · 2018-11-02
**Two-stage VAE method to generate high-quality samples and avoid blurriness**

**Rating:** 6
**Confidence:** 3

**Review:**

This paper proposed a two-stage VAE method to generate high-quality samples and avoid blurriness. It is accomplished by utilizing a VAE structure on the observation and latent variable separately. The paper exploited a collection of interesting properties of VAE and point out the problem existed in the generative process of VAE.  I have several concerns about the paper:

1.	It is necessary to explain why the second-stage VAE can have its latent variable more closely resemble N(u|0,I). Even if the latent variable closely resemble N(u|0,I), How does it make sure the generated images are realistic? I admit that the VAE model can reconstruct realistic data based on its inferred latent variable, however, when given a random sample from N(u|0,I), the generated images are not good, which is true when the dimension of the latent space is high. I still can’t understand why a second-stage VAE can relief this problem.
2.	The adversarial auto-encoder is also proposed to solve the latent space problem, by comparison, what is the advantage of this paper?
3.	Why do you set the model as two separate stages? Will it enhance the performance if we train theses two-stages all together?
4.	The proofs for the theory 2 and 3 are under the assumption that the manifold dimension of the observation is r, while in reality it is difficult to obtain this r, do these theories applicable if we choose a value for the dimension of the latent space that is smaller than the real manifold dimension of the observation? How will it affect the performance of the proposed method?
5.	The value of r and k in each experiment should be specified.

---

> ### Author Response · Authors · 2018-11-10
> **Response to AnonReviewer3 -- Part 2**
>
> 2.	Reviewer Comment:  The adversarial autoencoder is also proposed to solve the latent space problem, by comparison, what is the advantage of this paper?
>
> Our Response:  The adversarial autoencoder (Makhzani et al., 2016) requires adversarial training, meaning that like all GAN-related models, a complex min-max problem must be optimized in search of a saddle point.  A well-recognized advantage of VAEs is that the training involves pure minimization of a fixed variational energy function, which is generally more stable and resistant to mode collapse.  We should also point out that unlike VAEs, the adversarial autoencoder has no mechanism for pruning superfluous dimensions in the latent space.  Regardless of these key differences, we are aware of no published work where the adversarial autoencoder has been shown to produce competitive results generating novel samples like other GAN-related models (rather it has been tested on auxiliary tasks like semi-supervised learning, which is not in our scope).  Indeed the exhaustive recent testing from (Lucic et al., 2018) upon which we based our experiments, does not even include the adversarial autoencoder as a benchmark.
>
>
>
> 3.	Reviewer Comment:  Why train the model as two separate stages? Will it enhance the performance if we train these two stages together?
>
> Our Response:  We have addressed this question on the bottom of page 7, which states the following: "It should also be emphasized that concatenating the two stages and jointly training does not improve the performance. If trained jointly the few extra second-stage parameters are simply hijacked by the dominant objective from the first stage and forced to work on an incrementally better fit of the manifold. As expected then, on empirical tests (not shown) we have found that this does not improve upon standard VAE baselines."  Our theoretical results and algorithm development from Sections 2-4 also directly support this conclusion.  Regardless, we are happy to clarify further if needed.
>
>
>
> 4.	Reviewer Comment:  Do the technical proofs require knowledge of the ground-truth manifold dimensions r?  And how is the proposed algorithm affected when r is unknown?
>
> Our Response:  None of our proofs require that the ground-truth r is known explicitly in advance.  All that is required is that we set kappa >= r (please see proof statements for Theorems 1-3).  In other words, we only need to set the latent dimension kappa to be bigger than the ground-truth manifold dimension r.  The VAE then has a natural mechanism in place for discarding superfluous dimensions.  Of course obviously in practice if we set kappa to be far too large, then the training could potentially become a bit more difficult, since in addition to learning the correct ground-truth manifold, we are also burdening the model to detect a much larger number of unnecessary dimensions.  But the VAE is arguably more robust to kappa than most methods, and the basic point still holds:  we need not set kappa = r, we just need to choose kappa to be a reasonable value that is at least as big as r.  In contrast, if we set kappa < r, then the theory starts to break down and practical performance will begin to degrade as expected.
>
>
>
> 5.	Reviewer Comment:  The value of r and kappa in each experiment should be specified.
>
> Our Response:  The true latent manifold dimension r is unknown in all of our experiments since we are using real-world data.  However, for the dimension of the VAE latent code, we chose kappa = 64 for all experiments, except for the 2-Stage VAE* model results, where we used 32 for MNIST and Fashion-MNIST, 192 for CIFAR-10, and 256 for CelebA.  Note that these values were not carefully tuned and need not be exact per the arguments responding to reviewer comment 4 above.  We just tried a single smaller value for the simpler data (MNIST and FashionMNIST), and a couple larger values for the more complex ones (CIFAR-10 and CelebA).

---

> > ### Comment · AnonReviewer3 · 2018-11-26
> > **problem with question 3**
> >
> > Thanks for the detailed reply. The answer for question 3 still bothers me. The authors state that the joint training of the two stage have no benefit for the model. This does not make sense, and the reason cannot convince me. There are many popular hierarchical generative models, i.e. DBM[1], DBN[2], GBN[3], which have an enhanced performance in joint training. I think the authors should find out the reason for the failed joint training.
> >
> > [1] Salakhutdinov R, Larochelle H. Efficient learning of deep Boltzmann machines[C]//Proceedings of the thirteenth international conference on artificial intelligence and statistics. 2010: 693-700.
> > [2] Hinton G E. Deep belief networks[J]. Scholarpedia, 2009, 4(5): 5947.
> > [3] Zhou M, Cong Y, Chen B. Gamma Belief Networks[J]. arXiv preprint arXiv:1512.03081, 2015.

---

> > > ### Author Response · Authors · 2018-11-28
> > > **Detailed Explanation -- Part II**
> > >
> > > Returning to the original question, how might joint training of the first and second VAE stages interfere with this process?  The problem lies in the dominant influence of the reconstruction term from the first VAE stage.  As the decoder variance goes to zero (as needed for perfect reconstruction), this term can be pushed towards minus infinity at an increasingly fast rate.  If trained jointly, the extra degrees-of-freedom from the second-stage VAE parameters will be distracted from their original intended purpose of modeling q(z).  Instead they will largely be used to push the dominant reconstruction term even further towards minus infinity (with increasing marginal gains) at the expense of working to address criteria (ii) which has only a modest effect on the overall cost.
> > >
> > > Another way to think about this is to consider the following illustrative scenario.  Suppose we have a 2-stage VAE model that produces a reconstruction error that is infinitesimally close to zero, but provides a poor estimate of q(z).  Because the reconstruction term becomes increasingly dominant when close to zero per the analysis from Section 3, during joint training *all* parameters, including those from the second stage, will focus on pushing the reconstruction error even closer to zero, rather than improving the estimate of q(z).  But from a practical standpoint generating realistic samples this is unhelpful, because it is far better to improve the estimate of q(z) than to make the reconstruction error infinitesimally closer to zero.  This is why separate training is so critical, because it isolates the second-stage and forces it to address criteria (ii), rather than needlessly focusing on infinitesimal changes to the reconstruction term from criteria (i) that makes no perceptual difference to generated samples.  And indeed, when we do train jointly, although the reconstruction errors are quite small as expected, the more pivotal FID scores measuring sample quality are bad precisely because q(z) has been neglected.
> > >
> > > Regardless, we realize that there are many subtleties involved here, and hope that the above comments provide helpful clarification and background.

---

> > > ### Author Response · Authors · 2018-11-28
> > > **Detailed Explanation -- Part I**
> > >
> > > To help provide a clearer explanation of this phenomena, we revisit the two criteria required for producing good samples from a generative model built upon an autoencoder structure (like a VAE).  Per the analysis from reference (Makhzani et al., 2016) and elsewhere, these criteria are: (i) small reconstruction error when passing through the encoder-decoder networks, and (ii) an aggregate posterior q(z) that is close to some known distribution like N(0,I) that is easy to sample from.  As mentioned in a previous response, the latter criteria ensures that we have access to a tractable distribution from which we can easily generate random input samples that, when passed through the learned decoder, will be converted to output samples resembling the training data.
> > >
> > > The two stages of our proposed VAE model can be motivated in one-to-one correspondence with these two criteria. In brief, the first VAE stage addresses criteria (i) by pushing both the encoder and decoder variances towards zero such that accurate reconstruction is possible.  However, the detailed analysis from Sections 2 and 3 of our submission suggests that as these variances go towards zero to achieve this goal, the reconstruction cost dominates the overall VAE objective because the ambient space is higher-dimensional than the latent space where the KL penalty resides.  The consequence is that, although criteria (i) is satisfied, the aggregate posterior q(z) need not be close to N(0,I) (this is predicted by theory and explicitly confirmed by experiments, e.g., see Figure 1, rightmost plot).  This then implies that if we take samples from N(0,I) and pass them through the learned decoder, the result will not closely resemble samples from the training data.
> > >
> > > Of course if we had a way to directly sample from q(z), we would not need to use N(0,I), since by design of any autoencoder-structured generative model samples from q(z) passed through the decoder will represent the training data (assuming the reconstruction criteria has been satisfied as mentioned above).  Therefore, the second VAE stage of our proposal can be viewed as addressing criteria (ii) by learning a tractable approximation of q(z) that we can actually sample from intead of N(0,I).  This estimate of q(z) is formed from a special, independent VAE structure explicitly designed such that the ambient and latent spaces have the same dimension allowing us to apply Theorem 1, which guarantees that a good approximation can be found when reconstruction and KL terms are in some sense properly balanced.  Therefore, we now have access to a tractable process for producing samples from q(z), even though q(z) need not be close to N(0,I).  Per the notation of our submission on page 6, bullet point 3, sampling u from N(0,I) and then z from p(z|u) is a close approximation to sampling z from q(z).  This z can then be passed to the first-stage decoder to produce the desired data x.

---

> > > ### Author Response · Authors · 2018-11-28
> > > **High-Level Response**
> > >
> > > Thank you for reading our earlier response carefully and showing continued interest in understanding the details.  Just to clarify though, we are not arguing that joint training is unhelpful in other types of hierarchical generative models (such as in the references the reviewer mentioned, where we agree it can be advantageous).  Rather, our analysis merely suggests that within the narrow context of our particular 2-stage VAE structure, joint training is unlikely to be beneficial.  But the underlying reason for this is not actually a mystery.  Although admittedly counterintuitive at first, the inadequacy of joint training is exactly what is predicted by the theory (the same core analyses that inspired our non-obvious approach to begin with).   Furthermore, this prediction can be empirically tested, which we have done in multiple ways.  For example, we have tried fusing the respective encoders and decoders from the first and second stages to train what amounts to a slightly more complex single VAE model.  We have also tried merging the two stages including the associated penalty terms.  In both cases, joint training does not help at all, with performance no better than the first stage VAE (which contains the vast majority of parameters).

---

> ### Author Response · Authors · 2018-11-10
> **Response to AnonReviewer3 -- Part 1**
>
> Thanks for providing feedback regarding our submission and indicating specific points of uncertainty.  We provide detailed answers to each question as follows:
>
>
> 1.	Reviewer Comment:  Why do the second-stage VAE latent variables more closely resemble N(0,I), and how does this ensure that the generated samples are realistic, especially if the dimension of the latent space is high?
>
> Our Response:  These issues are addressed in Section 4 of our paper, building on foundational properties of VAE models and our theory from Sections 2 and 3, but we can provide some additional background details here.  First, it can be helpful to check reference (Makhzani et al., 2016) which defines the aggregate posterior q(z) = \int q(z|x)p_gt(x)dx, where q(z|x) serves as the encoder and p_gt(x) is the ground-truth data density.  The basic idea behind generative models framed upon an autoencoder structure (VAE or otherwise) is that two criteria are required for producing good samples: (i) small reconstruction error when passing through the encoder-decoder networks, and (ii) an aggregate posterior q(z) that is close to some known distribution like N(0,I) that is easy to sample from.  Without the latter criteria, we have no tractable way of generating random inputs to the learned decoder that will produce realistic samples resembling the training data distribution.
>
> In the context of our paper and VAE models, we argue that the first-stage VAE provides small reconstruction errors using a minimal number of latent dimensions (if parameterized properly with a trainable decoder variance), but not necessarily an aggregate posterior q(z) that is close to N(0,I).  This is because the basic VAE cost function is heavily biased towards finding low-dimensional manifolds upon which the data resides at the expense of learning the correct distribution within this manifold, which also prevents the aggregate posterior from nearing N(0,I).  However, although the VAE may partially fail in this regard, it nonetheless provides a useful mapping to a lower-dimensional space in such a way that we can apply Theorem 1 from our work.  In this lower dimensional space we treat q(z) \neq N(0,I) as a revised ground-truth data distribution p_gt(z), and train a new VAE with latent variables u.  Based on Theorem 1, in this restricted setting there will exist at least some parameterizations of the new encoder q(u|z) and decoder p(z|u) such that perfect reconstructions are possible, p_gt(z) is fully recovered, and KL[ q(u|z) || p(u|z) ] -> 0.  If this all occurs, then we have the new second-stage aggregate posterior
>
> q(u) = \int q(u|z)p_gt(z)dz  =  \int p(u|z)p_gt(z)dz = \int p_gt(z|u)p(u)dz  = p(u) = N(0,I)
>
> as desired.  For practical deployment, we then only need sample u from N(0,I), then z from p(z|u), and finally x from p(x|z).  Note also that if the latent dimension of z is higher than actually needed, the first-stage VAE decoder is effectively capable of blocking/pruning the extra dimensions as discussed in Section 3.  This will not guarantee high quality samples, but it is adequate for preparing the data from the aggregate posterior q(z) to satisfy Theorem 1, which can then be leveraged by the second-stage VAE as mentioned above and in our paper.

---

### Official Review · AnonReviewer2 · 2018-11-06
**The paper establishes a new state-of-art FID scores among auto-encoder based generative models with solid theoretical insights supporting the empirical results**

**Rating:** 9
**Confidence:** 4

**Review:**

The paper provides a number of novel interesting theoretical results on "vanilla" Gaussian Variational Auto-Encoders (VAEs) (sections 1, 2, and 3), which are then used to build a new algorithm called "2 stage VAEs" (Section 4). The resulting algorithm is as stable as VAEs to train (it is free of any sort of adversarial training, it comes with a little overhead in terms of extra parameters), while achieving a quality of samples which is *very impressive* for an Auto-Encoder (AE) based generative modeling techniques (Section 5). In particular, the method achieves FID score 24 on the CelebA dataset which is on par with the best GAN-based models as reported in [1], thus sufficiently reducing the gap between the generative quality of the GAN-based and AE-based models reported in the literature.

Main theoretical contributions:

1. In some cases the variational bound of Gaussian VAEs can get tight (Theorem 1).
In the context of vanilla Gaussian VAEs (Gaussian prior, encoders, and decoders) the authors show that if (a) the intrinsic data dimensionality r is equal to the data space dimensionality d and (b) the latent space dimensionality k is not smaller than r then there is a sequence of encoder-decoder pairs achieving the global minimum of the VAE objective and simultaneously (a) zeroing the variational gap and (b) precisely matching the true data distribution. In other words, in this setting the variational bound and the Gaussian model does not prevent the true data distribution from being recovered.

2. In other cases Gaussian VAEs may not recover the actual distribution, but they will recover the real manifold (Theorems 2, 3, 4 and discussions on page 5).
In case when r < d, that is when the data distribution is supported on a low dimensional smooth manifold in the input space, things are quite different. The authors show that there are still sequences of encoder-decoder pairs which achieves the global minimum of the VAE objective. However, this time only *some* of these sequences converge to the model which is in a way indistinguishable from the true data distribution (and thus again Gaussian VAEs do not fundamentally prevent the true distribution from being recovered). Nevertheless, all sequences mentioned above recover the true data manifold in that (a) the optimal encoder learns to use r dimensional linear subspace in the latent space to encode the inputs in a lossless and noise-free way, while filling the remaining k - r dimensions with a white Gaussian noise and (b) the decoder learns to ignore the k - r noisy dimensions and use the r "informative" dimensions to produce the outputs perfectly landing on the true data manifold.

Main algorithmic contributions:
(0) A simple 2 stage algorithm, where first a vanilla Gaussian VAE is trained on the input dataset and second a separate vanilla Gaussian VAE is trained to match the aggregate posterior obtained after the first stage. The authors support this algorithm with a reasonable theoretical argument based on theoretical insights listed above (see end of page 6 - beginning of page 7). The algorithm achieves state-of-art FID scores across several data sets among AE based models existing in the literature.

Review summary:
I would like to say that this paper was a breath of fresh air to me. I really liked how the authors make a strong point that *it is not the Gaussian assumptions that harm the performance of VAEs* in contrast to what is usually believed in the field nowadays. Also, I think *the reported FID scores alone may be considered as a significant enough contribution*, because to my knowledge this is the first paper significantly closing the gap between generative quality of GAN-based models and non-adversarial AE-based methods.

***************
*** Couple of comments and typos:
***************
(0) Is the code / checkpoints going to be available anytime soon?
(1) I would mention [2] which in a way used a very similar approach, where the aggregate posterior of the implicit generative model was modeled with a separate implicit generative model. Of course, two approaches are very different ([2] used an adversarial training to match the aggregate posterior), however I believe the paper is worth mentioning.
(2) In light of the discussion on page 6 as well as some of the conclusions regarding commonly reported blurriness of the VAE models, results of Section 4.1 of [3] look quite relevant.
(3) It would be nice to specify the dimensionality of the Sz matrix in definition 1.
(4) Line ater Eq. 3: I think it should be $\int p_gt(x) \log p_\theta(x) dx$ ?
(5) Eq 4: p_\theta(x|x)
(6) Page 4: "... mass to most all measurable...".
(7) Eq 34. Is it sqrt(\gamma_t) or just \gamma_t?
(8) Line after Eq 40. Why exactly D(u^*) is finite?

I only checked proofs of Theorems 1 and 2 in details and those looked correct.

[1] Lucic et al., 2018.
[2] Zhao et al., Adversarially regularized autoencoders, 2017, http://proceedings.mlr.press/v80/zhao18b.html
[3] Bousquet et al., From optimal transport to generative modeling: the VEGAN cookbook. 2017, https://arxiv.org/abs/1705.07642

---

> ### Author Response · Authors · 2018-11-10
> **Response to AnonReviewer2**
>
> We appreciate the detailed and positive comments, which truly reflect many of the essential contributions of our work.  Likewise to the best of our knowledge, the FID scores we report are indeed the first to close the gap between GANs and non-adversarial AE-based methods as the reviewer points out.  Regarding the small comments concluding the review, we answer as follows:
>
>
> - Reviewer Comment:  Is the code / checkpoints going to be available anytime soon?
>
> Our Response: It was our original intention to simply post the code on Github after decisions were issued and papers were de-anonymized.  However, if there is a need to make the code available earlier while preserving anonymity, we could presumably pursue that as well (but not sure if this is considered acceptable under ICLR guidelines).
>
>
> - Reviewer Comment:  Reference to an alternative method for estimating the aggregate posterior, and another paper addressing causes of blurry VAE representations.
>
> Our Response:  Thanks for the nice references.  These papers actually look very interesting; we can cite them and provide context in the revision.
>
>
> - Reviewer Comment: Line after Eq. 3: I think it should be \int p_gt(x) \log p_\theta(x) dx ?
>
> Our Response: It is true that L(\theta, \phi) >= - \int p_gt(x) \log p_\theta(x) dx. However, we further have that - \int p_gt(x) \log p_\theta(x) dx  >=  -\int p_gt(x) \log p_gt(x) dx, which is the expression we include below Eq. (3) in the paper. The equality holds iff KL[q_\phi(z|x) || p_\theta(z|x)] = 0 and p_\theta(x) = p_gt(x) almost everywhere.
>
>
> - Reviewer Comment: Line after Eq 40. Why exactly is D(u^*) finite?
>
> Our Response: Because \varphi(u) is a diffeomorphism, is has a differentiable inverse and \Lambda(u) = (d\varphi(u)^-1/du)^\top (d\varphi(u)^-1/du) is always finite. Furthermore, D(u^*) is the maximum of \Lambda(u) in a closed set centered at u^*, so it is finite.  We will update the proof to include these extra details.
>
>
> - Reviewer Comment: Minor typos/corrections
>
> Our Response:  Thanks for catching each of these and also checking the proofs carefully.  We have fixed each typo/suggestion in a revised version.

---

### Public Comment · (anonymous) · 2018-11-12
**Relationship with the Vamp prior**

The two-stage process you introduce seems very closely related to using a Vamp prior (https://arxiv.org/abs/1705.07120), wherein one effectively tries to replace the original prior with the aggregate posterior q(z) (though this is not achieved exactly for computational reasons).  Obviously, there are some differences, but this seems like a natural baseline that should probably be compared to and at the very least a paper that should be being cited and discussed.

---

> ### Author Response · Authors · 2018-11-14
> **Response to the Vamp prior reference suggestion**
>
> Thank you for the reference to (Tomczak and Welling, 2018), which proposes a nice two-stage hierarchical prior to replace the parameter-free standardized Gaussian N(0,I) that is commonly used with VAE models.  Note that multiple stages of latent variables have been aggregated in the context of VAE-like models going back to (Rezende et al., 2014).  However, beyond the common use of two sets/stages of latent variables, our approach bares relatively little similarity to (Tomczak and Welling, 2018) or other multi-stage alternatives.   For example, the underlying theoretical design principles/analysis, aggregate energy function parameterizations, and training strategies are not at all the same.  Likewise, the empirical validation is completely different and incomparable as well; (Tomczak and Welling, 2018) focuses on demonstrating improved log-likelihood scores, while we concentrate exclusively on improving the quality of generated samples as explicitly quantified by FID scores.  And we stress that these two evaluation criteria can be almost completely unrelated to one another in many circumstances (see for example, Theis et al., "A Note on the Evaluation of Generative Models," ICLR 2016).  And as a final point of differentiation, (Tomczak and Welling, 2018) tests only on small black-and-white images and includes no comparisons against GANs, while we include testing with larger color images like CelebA and directly compare against GANs in a neutral setting.  Regardless, (Tomczak and Welling, 2018) still represents a compelling contribution, and space permitting, we can try to provide broader context in a revision.

---

> > ### Public Comment · (anonymous) · 2018-11-14
> > **Link is much stronger and more subtle than this**
> >
> > Thanks for your reply.  I appreciate that there are certainly differences between the two, including in their original motivations, and I certainly not trying to imply your work is just a rehashing of theirs.  I should point out that I am in no way associated with that paper so I have no ulterior motive to try and promote it or similar.
> >
> > However, I think the link between the two is a lot stronger than something to do with hierarchical priors and so I disagree with your suggestion above.  The link is that both consider issues caused by the mismatch between the aggregate posterior q(z) and the prior p(z).  In your work, you learn a second network to generate samples from q(z) and thus in turn p(x|z)q(z).  In their formulation, they instead replace p(z) with q(z), therefore generating samples from exactly the same model as yours, at least in theory.  In practice, they have to make approximations because q(z) is not directly available.  Consequently, the two approaches are intimately linked to one another, the key methodological differences, in my opinion, being that in your case you only approximate q(z) after training and you use a different method to approximate q(z).  There is a bit of a trade-off here, your method for approximating q(z) is almost certainly better, but this better approximation prevents you using it during training, which is likely to lead to a worse model being learned.
> >
> > Consequently, I think the link is a lot stronger than you are suggesting above, and thus this is an essential piece of related work to be considering.

---

> > > ### Author Response · Authors · 2018-11-17
> > > **Response**
> > >
> > > Thanks for the continued engaging dialogue, and we can try to further clarify what we believe to be critical differentiating factors.  First, you mentioned that the link between our method and (Tomczak and Welling, 2018) is that we both consider issues caused by the mismatch between the aggregate posterior q(z) and the prior p(z).  But whether explicitly stated or not, essentially all methods based on an autoencoder structure share this exact same link on some level, so this is not any particular indication of close kinship in and of itself.  And if this mismatch is ignored, then samples drawn from p(z) and passed through the decoder are highly unlikely to follow the true ground-truth distribution (see for example (Makhzani et al., 2016) mentioned in our submission).
> > >
> > > Beyond this though, the means by which we deal with this central, shared issue are fundamentally different.  In our case, we exploit provable conditions whereby an independent second-stage VAE can effectively learn and sample from the unknown q(z) produced by a first stage VAE, and additionally, we provide direct empirical evidence supporting this theory (e.g., see Figure 1, righthand plot).  Hence it no longer matters that p(z) and q(z) are not the same since we can just sample from the latter using the second-stage VAE.  Even though this approach may seem counter-intuitive at first glance, an accurate model can in fact be learned (provably so in certain situations), and our state-of-the-art results for a VAE model relative to GANs (the very first such demonstration in the literature) provide further strong corroborating evidence.
> > >
> > > In contrast, (Tomczak and Welling, 2018) choose to parameterize p(z) in such a way that the additional flexibility can provide simpler pathways for pushing p(z) and q(z) closer together.  This is certainly an interesting idea, but it is significantly different from ours.  But of course we agree that the ultimate purpose is the same:  to have access to a known distribution with which to initiate passing samples through the decoder, a common goal shared by all autoencoder-structured models, including ours and many others like (Makhzani et al., 2016), where an adversarial loss is used to push p(z) and q(z) together.  What ultimately distinguishes these methods is, to a large degree, the specific way in which this goal is addressed.  We have no reservations about including additional discussion of (Tomczak and Welling, 2018), and these broader points in a revised version of our paper.

---

> > > > ### Public Comment · (anonymous) · 2018-11-18
> > > > **Thanks**
> > > >
> > > > Thanks for the reply.  I think this is a great exposition of the differences and the paper will be strengthened by making some of these points in the revision.

---

### Public Comment · (anonymous) · 2018-11-13
**A question regarding the analysis on Eq. (9) and the 2-stage VAE**

It is an interesting and refreshing paper. I have a question regarding the analysis on Eq. (9). When \gamma->0, the coefficient (1/\gamma) of the reconstruction term of Eq. (9) will approach infinity, which results in a loss function that is similar to that of a plain AE. To see that, we can multiply Eqs. (8) and (9) by \gamma, then the coefficient of the reconstruction term becomes 1, while that of Eq. (9) approaches 0. Note that \gamma\log(\gamma)->0 when \gamma->0. So I don't see why \hat{r} will be pushed to as small as possible. Intuitively, if we add some small (e.g., stddev=0.01) isotropic gaussian noise to x, we wouldn't expect the resulting model to be significantly different, while the analysis seems to suggest that \hat{r} will suddenly increase from r to \kappa (assuming \kappa<d), since the manifold of the noisy x is d-dimensional. Moreover, it would be interesting to see if adding a second stage VAE on top of a plain AE can lead to similar performance gain.

---

> ### Author Response · Authors · 2018-11-14
> **Response to the question**
>
> Thanks for your interest in our work.  Regarding the situation when gamma -> 0, the VAE will not actually default to a regular AE.  Note that we can multiply both reconstruction and regularization terms (eqs. (8) and (9)) by gamma and then examine the limit as gamma becomes small; however, this does not allow us to discount all of the regularization factors even though they may be converging to zero as well.  The convergence details turn out to be critical here.
>
> To see this, consider the following simplified regularized regression problem which reflects the core underlying issue.  Assume that we would like to solve
>
> min_w (1/gamma)||y - A w||^2 + ||w||^2,
>
> where 1/gamma is a trade-off parameter, y is a known observation vector, A is an overcomplete matrix (full rank, with more columns than rows), and w represents unknown coefficients we would like to compute.  If gamma -> 0, then any optimal solution must be in the feasible region where y = A w, meaning zero reconstruction error.  Therefore, when gamma -> 0 solving this problem becomes formally equivalent to solving the constrained problem
>
> min_w ||w||^2  subject to y = A w.
>
> Of course we could equally well consider multiplying both sides of the original objective by gamma, producing
>
> min_w ||y - A w||^2 + gamma ||w||^2.
>
> This shouldn't change the optimal w since we have just multiplied by a constant independent of w.  But if gamma -> 0, then technically speaking, the regularization factor gamma ||w||^2 becomes arbitrarily small; however, this does not mean that we can simply ignore it because there are an infinite number of solutions whereby the data factor ||y - A w||^2 equals zero, i.e., a fixed, minimizing constant.  The direct implication is that
>
> limit gamma -> 0 arg min_w ||y - A w||^2 + gamma ||w||^2  \neq  arg min_w ||y - A w||^2,
>
> where the righthand side is just the objective obtained when gamma = 0, and it has an infinite number of minimizers unlike the lefthand side.  In general, the regularization factor ||w||^2 will always have an influence in choosing which solution, out of the infinite number satisfying y = A w, is optimal, and the minimizing argument will again provably be the same as from the constrained problem above.  This notion is well-established in the regularized regression literature, and generalizes to generic problems composed of data-fitting and regularization terms where the former in isolation has multiple equivalent minima.  Returning to the VAE, if extra unneeded latent dimensions are present, then there will be an infinite number of latent representations capable of producing perfect reconstructions.  The lingering KL regularization terms then determine which is optimal, per our analysis in Section 3 of the paper.
>
> Additionally, in terms of adding small isotropic noise to observations x, the results will actually not be much different. This is because in practice, gamma need not converge to exactly zero, but only a small value near zero.  This allows the model to slightly expand around the manifold and still apply high probability to the data.  If the noise level is within such a modest expansion, then the behavior is more-or-less the same as if a low-dimensional manifold were present. Of course if added noise or other deviations from the manifold are too large, then obviously using additional dimensions to model the data may be required.
>
> Finally, with regard to your other question, we have also considered training a second-stage VAE on top of a regular autoencoder.  This structure is discussed in footnote 5 on page 7.

---

### Author Response · Authors · 2018-11-30
**Update regarding empirical evaluations**

After our original submission, we have continued investigating a wider variety of generative models and evaluation metrics for broader research purposes.  We summarize a few updates here that are relevant to our submission:

* As a highly-relevant benchmark, we have obtained additional FID scores for all of the GAN models trained using suggested hyperparameter settings (from original authors), as opposed to the scores we originally reported from (Lucic et al., 2018) that were based on a large-scale, dataset-dependent hyperparameter search.  When averaged across all four datasets (i.e., MNIST, Fashion, CIFAR10, CelebA), all GAN models trained with suggested settings had a mean FID score above 45.  In contrast, with hyperparameters optimized across 100 different settings independently for each dataset as in (Lucic et al., 2018), the mean GAN FID scores are all within the range 31-45.  As a point of reference, our proposed 2-Stage VAE model with no tuning whatsoever (the same default settings across all datasets) has a mean FID below 40, which is significantly better than all of the GANs operating with analogous fixed/suggested settings, and well within the range of the heavily-optimized GANs.  And all other existing VAE baselines we have tried (including additional ones computed since our original submission), are considerably above this range.

* In our original submission we also included results from a model labeled 2-Stage VAE*, where we coarsely optimized the hyperparameter kappa (the dimension of the latent representation).  However, upon further reflection we have decided that it is probably better to remove this variant for two reasons.  First, although the optimized GAN models involved searching over values from 7 hyperparameter categories (see the supplementary file from the latest NeurIPS 2018 version of (Lucic et al., 2018)), varying kappa was apparently not considered.  Therefore it is somewhat of an apples-and-oranges comparison between our 2-Stage VAE* and the optimized GANs.  Secondly, we have recently noticed that PyTorch and TensorFlow implementations of FID scores are sometimes a bit different (this appears to be the result of different underlying Inception models upon which the FID score is based).  This discrepancy is inconsequential for our 2-Stage VAE model and associated baselines, but for 2-Stage VAE* the mean improvement differs by 4 depending on the FID implementation (this could be in part because optimizing over FID scores may exacerbate implementation differences).  Regardless, this issue highlights the importance of using a consistent FID implementation across all models (a seemingly under-appreciated issue in the literature).

* Although normalizing flows have been frequently reported to improve log-likelihood values in VAE models, this type of encoder enhancement has not as of yet been shown to improve FID scores (at least in the literature we are aware of).  Of course log-likelihood values are not a good indicator of generated sample quality at measured by FID (Theis et al., ICLR 2016), so improving one need not correlate with improving the other.  Even so, per the suggestion of AnonReviewer1, we have conducted experiments using VAE models with normalizing flows (Rezende and Mohamed, ICML 2015) as an additional baseline.  Thus far, we have not found any instances where the addition of flows improves the FID score within the standardized/neutral testing framework from (Lucic et al., 2018), and sometimes the flows can actually make the FID worse.  Still there are numerous different flow-based models, and further investigation is warranted to examine whether or not some versions could indeed help in certain scenarios.

* Finally, we have also performed evaluations using the new Kernel Inception Distance (KID) quantitative metric of sample quality.  This metric was proposed in (Binkowski et al., ICLR 2018) and serves as an alternative to FID.  Note that we cannot evaluate all of the GAN baselines using the KID score; only the authors of (Lucic et al., 2018) could easily do this given the huge number of trained models involved that are not publicly available, and the need to retrain selected models multiple times to produce new average scores at optimal hyperparameter settings.  However, we can at least compare our trained 2-Stage VAE model to other VAE baselines.  In this regard we have found that the same improvement patterns reported in our original submission with respect to FID are preserved when we apply KID instead, providing further confidence in our approach.

---

### Public Comment · (anonymous) · 2019-03-25
**Clarification on reported FID values**

Sorry for the late comment. I am trying to reproduce your results using your code at https://github.com/daib13/TwoStageVAE

I set the network structure to "Infogan", as that is what you reported using for the results in Table 1 of your paper. In your code it seems that a learnable gamma is the default setting and is not configurable. After training on CIFAR-10, I observed FID scores of 101.55 and 100.01 for the VAE and 2-stage VAE models, respectively. These are far from the mean values of 76.7 and 72.9 reported in Table 1.

I was just wondering if you could clarify that "Infogan" is the correct setting for the network structure to reproduce these results? Thanks

---

> ### Public Comment · (anonymous) · 2019-03-25
> **Oh**
>
> Allow me to answer my own question: The default number of trained epochs for the code is 400 while in the paper the reported epochs trained for CIFAR-10 is 1000. I will delete my comment and retrain.

---

> > ### Public Comment · (anonymous) · 2019-03-25
> > **After retraining...**
> >
> > I'm still getting a massive discrepancy between the FID scores reported in the paper and the FID scores I'm seeing after running the code.
> >
> > For reference this is the command I'm using to train the model:
> >
> > python demo.py --dataset cifar10 --network-structure Infogan --epochs 1000 --epochs2 2000 --lr-epochs 300 --lr-epochs2 600 --batch-size 100
> >
> > FID scores are 100.58 and 94.93 for the VAE and 2-stage VAE models respectively. Any clarification would be appreciated, thanks.

---

> > > ### Author Response · Authors · 2019-04-03
> > > **Clarification**
> > >
> > > Sorry we responded to this last week, but just noticed that the response was located after your original comment (not the latest) and seemingly not accessible to everyone.  Anyway, the answer to your question is as follows:
> > >
> > > When we compute FID scores, we first save the generated images to a folder using ``scipy.misc.imsave'' and then read the images to calculate the required inception feature. This is slightly different than directly using the output of the decoder because imsave will automatically rescale images to use the full (0, 255) range.  This rescaling actually makes little difference to the FID scores on most datasets; however, for some reason CIFAR-10 data seems to be more sensitive, even though perceptually the generated images look the same.  In general though, FID score computations are potentially sensitive to seemingly inconsequential factors, such as whether the tensorflow or pytorch inception network is used, so it is important to use a consistent methodology across different methods.
> > >
> > > In the present case, to obtain results consistent with ours, generated images should be normalized to use the full (0,255) range as is done when using imsave under default settings.  Although it is difficult to know for sure, given the common use of the imsave function, it is likely that many other works handle FID computations in the same way.  We will update our github shortly to make these details more explicit.

---

### Meta-Review · Area_Chair1 · 2018-12-16
**interesting analysis of Gaussian VAEs, a simple VAE training approach that results in impressive sample quality**

**Confidence:** 4
**Recommendation:** Accept (Poster)

**Metareview:**

The reviewers acknowledge the value of the careful analysis of Gaussian encoder/decoder VAE presented in the paper. The proposed algorithm shows impressive FID scores that are comparable to those obtained by state of the art GANs. The paper will be a valuable addition to the ICLR program.